# Coverage evaluation surveys following soil-transmitted helminthiasis and schistosomiasis mass drug administration in Wolaita Zone of Ethiopia—The Geshiyaro project

Ewnetu Firdawek Liyew[1]*, Melkie Chernet[1], Habtamu Belay[1], Rosie Maddren[2,3], Toby Landeryou[2,3], Suprabhath Kalahasti[2], Alison K. Ower[2,3], Kalkidan Mekete[1], Anna E. Phillips[2,3], Ufaysa Anjulo[4], Tujuba Endrias[1], Adugna Tamiru[5], Bokretsion Gidey[1], Zelalem Mehari[6], Birhan Mengistu[6], Getachew Tollera[1], Geremew Tasew[1]

1 Bacterial, Parasitic and Zoonotic Diseases Research Directorate, Ethiopian Public Health Institute, Addis Ababa, Ethiopia, 2 Department of Infectious Disease Epidemiology, School of Public Health, Faculty of Medicine, Imperial College London, London, United Kingdom, 3 London Centre for Neglected Tropical Disease Research, London, United Kingdom, 4 Disease Prevention and Health Promotion Core Process, Ministry of Health, Wolaita, Ethiopia, 5 Disease Prevention and Control Directorate, Ministry of Health, Addis Ababa, Ethiopia, 6 Children's Investment Fund Foundation, London, United Kingdom

* ewnetuliyew@gmail.com

**Data Availability Statement:** All relevant data are within the paper and its Supporting information files.

## Abstract

### Introduction

The Geshiyaro project aims to break transmission of soil-transmitted helminths and schistosomiasis in the Wolaita Zone of Ethiopia through a combination of two interventions: behavior change communication (BCC) for increased water, sanitation and hygiene (WaSH) infrastructure use alongside preventive chemotherapy (PC) using albendazole (ALB) and praziquantel (PZQ), targeted to reach 90% treatment coverage. Coverage evaluation surveys (CES) were conducted post-treatment, and the resultant survey coverage was compared to reported administrative coverage. This provided a secondary confirmation of the Geshiyaro project coverages, and is used to monitor the success of each Mass Drug Administration (MDA) round.

### Methods

A community-based cross-sectional study was conducted in 13 woredas (districts) of the Wolaita Zone. All eligible individuals from the selected households were invited for an interview. The study design, sample size, analysis and report writing were conducted according to the World Health Organization (WHO) CES guidelines for PC.

### Results

The study interviewed a total of 3,568 households and 18,875 individuals across 13 woredas in the Wolaita Zone. Overall, the survey coverage across all studied woredas was 81.5% (95% CI; 80.9–82.0%) for both ALB and PZQ. Reported administrative coverage across all

**Funding:** This research was funded by the Children's Investment Fund Foundation (CIFF), UK through a grant to Ethiopian Public Health Institute, Ethiopia under grant number R-1805-02741. The funders had no role in the design of the study, collection, analysis and interpretation of the data. The finding and conclusion of the study reflect the view of the Authors only.

**Competing interests:** I have read the journal's policy and the authors of this manuscript have the following competing interests: ZM is the Manager, Evidence Measurement and Evaluation at the Children's Investment Fund Foundation. BM is the Program Manager at the Children's Investment Fund Foundation, the project's funder. This does not alter our adherence to PLOS ONE policies on sharing data and materials.

studied woredas was higher than survey coverage, 92.7% and 91.2% for ALB and PZQ, respectively. A significant portion of individuals (17.6%) were not offered PC. The predominant reason for not achieving the target coverage of 90% was beneficiary absenteeism during MDA (6.6% ALB, 6.8% PZQ), followed by drug distributors failing to reach all households (4.7% ALB, 4.8% PZQ), and beneficiaries not informed of the program (1.3% ALB, 1.7% PZQ).

## Conclusion

Programmatic actions will need to be taken during the next MDA campaign to achieve the targeted Geshiyaro project coverage threshold across data collection and program engagement. Adequate training and supervision on recording and reporting administrative coverage should be provided, alongside improved social mobilization of treated communities to increase participation, and strengthened institutional partnerships and communication.

## Introduction

Neglected tropical diseases (NTDs) are a diverse group of infectious diseases affecting one billion people globally [1, 2]. Two of the twenty recognised NTDs, soil-transmitted helminths (STH) and schistosomiasis (SCH), disproportionally affect those that live in poverty due to an inadequate sanitation and hygiene infrastructure [3, 4]. In sub-Saharan Africa, Ethiopia has the 5th highest STH prevalence, and 14th highest SCH prevalence [5]. In Ethiopia, it is estimated that 37.3 million and 79 million people live in SCH and STH endemic areas, respectively [6, 7]. The WHO recommends PC with ALB and PZQ to control STH and SCH, respectively, using either annual or bi-annual treatment intervals, proportional to infection prevalence [8–13].

The Geshiyaro project is designed to break transmission of STH and SCH, conducted over a period of five-years in the Wolaita Zone of Ethiopia. The project will measure the impact of a combination of two interventions; expanded community-wide MDA and the building of WaSH facilities with BCC, with the aim to inform potential endgame, elimination strategies for STH and SCH. The protocol for the project has been explained previously by Mekete et al. [14]. MDA and WaSH activities are organized and overseen by the Ministry of Health (MoH) and World Vision, respectively. Since 2018, in Geshiyaro project woredas of Wolaita, eligible community has been treated bi-annual ALB and an annual PZQ treatment, with the goal of reaching 90% treatment coverage at each MDA round. The kebele-level network of Health Extension Workers (HEW) were used to distribute the MDA, supported by Ethiopian Public Health Institute (EPHI) and MoH representatives [14].

The progress of each MDA round, is monitored by two indices; program reach and survey coverage. In this study we will refer to program reach as the percentage of the eligible population contacted, and survey coverage as the percentage of these eligible individuals who swallowed the drugs [12, 13]. Without reliable information about PC coverage it is a challenge to evaluate programme performance effectively, or indeed predict how the prevalence of infection and associated disease is impacted by the MDA [13]. The eligible population was calculated according to WHO guidelines and is drug specific: individuals aged 1 and older are eligible for ALB, whilst individuals older than 4 years are eligible for PZQ. Mothers in their first trimester are advised not to take ALB or PZQ [9, 11]. All eligible community members

were offered one dose of 400mg ALB (> 2 years old), one bottle 10ml syrup of 200 mg per 5ml ALB (1–2 years old) and 600mg PZQ (>4 years old) administered in a height-dependent dose (1–5 tablets).

Reported administrative PC coverage data calculated from drug distributor's handwritten records is important for programme monitoring, yet it is prone to errors resulting from incorrect estimates of the target population and therefore the denominator, weak health information systems, underreporting, or intentional inflation of individuals treated [13]. CES are population-based surveys designed to provide precise statistical estimate of the PC coverage that overcome many of the biases and errors that can undermine reported administrative coverage [13]. This makes the implementation of CES in the Geshiyaro project a valuable tool for evaluating program performance, and comparing the reported administrative coverage by drug distributors.

This study reported on the latest round of Geshiyaro expanded community-wide MDA, distributed in 2021 to 13 woredas in Wolaita. The estimated survey coverage taken from the CES reports is compared with the reported administrative coverage. The reasons given by the community for not participating in MDA is also assessed. The findings of this study will be important for the national program as the lessons learnt can be implemented to improve future MDA campaigns.

## Methods

### Study settings and period

This post-treatment CES was conducted in February 2021 in a random sample of the population from 13 woredas of Wolaita that received treatment. Wolaita is located in the south west of the Southern Nations and Nationalities Peoples Region (SNNPR), 330Km from Addis Ababa. According to the recent government restructure, Wolaita's original 15 woredas have been redistricted into 22 woredas. This study considers the latest round of MDA, administered in January 2021 across thirteen woredas of Wolaita.

### Study design

A community-based cross-sectional study design was used for the current study, taken from the larger five-year longitudinal study conducted for the Geshiyaro project.

### Sampling

For this CES, we designed it using probability proportional to estimated size (PPES) in selecting enumeration areas (EAs), the smallest administrative unit used in Ethiopian districting, from 13 survey woredas. An exhaustive list of EAs and the estimated number of households (HHs) for the respective woredas in Wolaita zone was obtained from the Ethiopian Central Statistics Authority (CSA). From each woreda, 30 EAs were randomly selected with a probability proportional to the segments of the EAs, whereby a segment represents a group of roughly 50 households (HHs) [13]. The Coverage Survey Builder (CSB), an Excel-based tool recommended by WHO [13], was used to select the EAs from each woreda. Following EA selection, a segment was selected randomly from each EA, and in turn HHs to be included in the study were also randomly selected. The sampling interval (the interval between two selected HHs), was automatically generated by the CSB, and used to determine which HHs in the segment were to be sampled in order to reach the required sample size. All eligible household members living in the selected HH were interviewed. The latest population-based mini-survey in Ethiopia was done in 2019 and was used to estimate the total number of households (HHs) for the

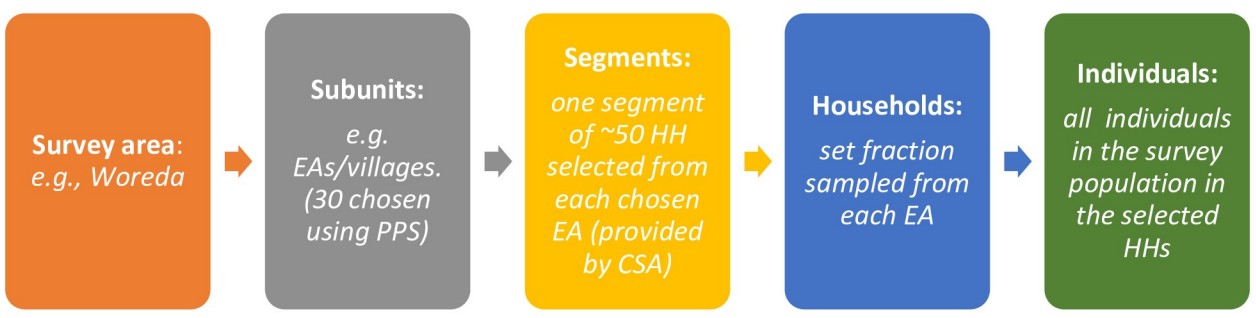

**Fig 1. Sampling scheme at different stages, adapted from the WHO [13].**

respective woredas [15]. Fig 1 shows the summary of sampling scheme used to select study participants.

## Study population

According to the Geshiyaro MDA program, ALB and PZQ had been provided to all individuals ($\geq$ 1 and >4 years old, respectively) residing in the Wolaita Zone. Therefore, the study population consists of all community members residing in 13 woredas who are eligible for ALB and PZQ drugs. PC survey coverage was estimated from the eligible study population which was considered as the denominator for the calculation of survey coverage.

## Sample size calculations

The sample size was determined automatically using the WHO Excel-based CSB [13] by assuming 0.05 margin of error, a 95% confidence interval (CI), non-response rate of 15%, and a design effect of 4 with an expected coverage rate taken from the reported administrative coverage by drug distributors (HEWs) during the MDA. The expected administrative coverage along with the detailed parameter used to calculate the sample size for each woreda is detailed in S1 Appendix.

## Data collection

Data was collected by well-trained health professionals using the Android smartphone Survey CTO application (Dobility, Inc; Cambridge, MA, USA) installed in each data collector's project mobile phone. The questionnaire used was adapted from the WHO CES tool [13], designed to capture pertinent information from the participants (S2 Appendix). The data collectors along with local kebele guides identified a walking route that passed every house in the selected segment. The adjacent HH was considered if the selected HH via the walking route was a business center. Pertinent information was obtained from all eligible household members living in the selected HH. Information on young children (<10 years) were collected from their primary caretakers. A "mop-up" activity took place for HHs whereby members were not present during the original survey activity. During the mop-up, HHs were revisited, and if the members were still absent, available adults answered the survey on behalf of the absent members. Interviews continued until the required sample size was obtained in each segment.

## Data quality control

To ensure high quality data collection, four days of intensive training was provided to data collectors and their supervisors covering; the usage of Survey CTO mobile phone application,

sampling methods employed, and questionnaire content. The WHO CES tool was adopted for the current study which avoided potential measurement errors. Additionally, the recommended WHO coverage evaluation protocol for PC was adapted to inform the implemented methods and reporting of this study [13]. Daily reports for data quality management were communicated to supervisors and data collectors, with the aim to update progress, and identify any errors to be rectified. To avoid recall bias, data collection was completed within one month of the MDA campaign. To avoid social desirability bias, the HEWs who originally distributed drugs during MDA were not used as local kebele guides. Additionally, the data collectors presented sample ALB and PZQ pills to aid recollection by HH members.

## Data analysis

Two percentage metrics were calculated in this study: the self-reported 'survey coverage', of what percentage of the eligible population swallowed a pill (which is used to compare with the administrative reported coverage) and the 'program reach' (which tell us whether the individual is offered the drug or not). Individual compliance with the MDA treatment was identified by comparing the survey coverage to the programme reach. Self-reported survey coverage was compared with the MoH-reported administrative coverage and the target Geshiyaro coverage threshold. The following formulas were used to calculate the survey coverage and program reach, respectively.

$$Survey\ coverage = \frac{Number\ of\ individuals\ who\ \textbf{swallowed}\ the\ drug}{Total\ number\ of\ individuals\ surveyed}$$

$$Program\ reach = \frac{Number\ of\ 'yes'\ responses\ to\ having\ been\ \textbf{offered}\ the\ drug}{Total\ number\ of\ individuals\ surveyed}$$

The WHO CSB "Results Entry Form" available in the CSB [13] was utilized for estimations of survey and program reach coverages. The 95% CI around the survey coverage was also automatically calculated using the CSB analysis tool.

## Ethics statement

The study was approved by Institutional Review Board (IRB) at the Scientific and Ethical Review Office of the Ethiopian Public Health Institute. A letter of support and explanation of the study purpose was provided to all relevant governmental bodies. For all non-experimental studies, obtaining a verbal consent is the standard requirement of the Institutional Review Board of Ethiopian Public Health Institute (EPHI). Hence, verbal consent was taken from HH after providing a summary of the study purpose in the local dialect, Wolaitigna. Assent for young children (<18 years) was obtained from their primary guardian. The confidentiality of all the participants was kept through the use of encrypted datasets, and individual's identification numbers linking demographic and MDA information.

## Results

### Total number of households and individuals interviewed

As seen in Table 1, a total of 3,568 HHs and 18,875 individuals were interviewed across the 13 studied woredas in Wolaita. Surveyed study participant gender was evenly distributed. School-aged children (SAC) (aged 5 to14 years) and pre-school aged children (pre-SAC) (aged 1 to 4 years) were over and under sampled, respectively.

**Table 1. Distribution of interviewed HHs and individuals across each woreda, Wolaita Zone, February 2021.**

| Woreda | Total HHs interviewed | Total individuals interviewed | Female | Male | Age group (year) | | | | |
|---|---|---|---|---|---|---|---|---|---|
| | | | | | 1–4 | 5–14 | 15–20 | 21–35 | 35+ |
| Boloso Sore | 251 | 1,391 | 663 | 728 | 142 | 449 | 238 | 332 | 230 |
| Sodo Town | 272 | 1,415 | 726 | 689 | 96 | 361 | 274 | 404 | 280 |
| Diguna Fango | 269 | 1,445 | 707 | 738 | 129 | 440 | 255 | 362 | 259 |
| Abala Abaya | 314 | 1,701 | 833 | 868 | 158 | 576 | 305 | 347 | 315 |
| Kindo Koysha | 288 | 1,652 | 851 | 801 | 125 | 530 | 300 | 431 | 266 |
| Offa | 268 | 1,432 | 727 | 705 | 150 | 437 | 216 | 299 | 330 |
| Sodo Zuria | 273 | 1,398 | 700 | 698 | 105 | 429 | 250 | 287 | 327 |
| Tebela Town | 270 | 1,364 | 687 | 677 | 108 | 346 | 287 | 364 | 259 |
| Humbo Woreda | 279 | 1,525 | 738 | 787 | 115 | 512 | 259 | 311 | 328 |
| Hobicha | 269 | 1,543 | 761 | 782 | 124 | 518 | 287 | 312 | 302 |
| Bayra Koysha | 276 | 1,260 | 639 | 621 | 101 | 375 | 182 | 295 | 307 |
| Kawo Koysha | 273 | 1,401 | 718 | 683 | 137 | 462 | 201 | 262 | 339 |
| Gesuba Town | 266 | 1,348 | 674 | 674 | 90 | 372 | 271 | 320 | 295 |
| **Total** | **3,568** | **18,875** | **9,424** | **9,451** | **1,580** | **5,807** | **3,325** | **4,326** | **3,837** |

## Survey coverage of albendazole and praziquantel by woreda

The overall survey coverage for both ALB and PZQ in the studied woredas was 81.5% (95% CI; 80.9–82.0%), shown in Tables 2 and 3. The lowest survey coverage for both ALB and PZQ were observed in Sodo Town, reaching only 52.3% and 51.3%, respectively. Conversely, high survey coverage was reported in Abala Abaya, Sodo Zuria and Bayra Koysha woredas. Five of 13 woredas reported a survey coverage above 85% for both drugs (Tables 2 and 3).

The overall survey coverage among male and female individuals for ALB were 81.8% and 81.1%, respectively, whilst the survey coverage for PZQ among male and female individuals were 81.1% and 81.9%, respectively. Generally, the overall survey coverage of both ALB and PZQ among males and females were not statistically different (*p = 0.9* for ALB and PZQ). The highest survey coverage for ALB and PZQ was observed in SAC at 87.6% for ALB and 86.4%

**Table 2. Comparison of survey coverages with reported administrative coverages, program reach and the Geshiyaro threshold (which is ≥90%) for ALB, Wolaita Zone, February 2021.**

| Woreda | Reported coverage (%) | Survey coverage with 95% CI | Program reach | Geshiyaro threshold | Drug acceptance (%) |
|---|---|---|---|---|---|
| Boloso Sore | 93.6 | 81.9 (74.7, 87.5) | 82.8 | 90 | 99.1 |
| Sodo Town | 83.5 | 52.3 (44.7, 59.8) | 53.2 | 90 | 98.3 |
| Diguna Fango | 97.4 | 83.8 (78.3, 88.1) | 84.7 | 90 | 98.9 |
| Abala Abaya | 83.1 | 88.2 (84.0,91.4) | 88.9 | 90 | 99.2 |
| Kindo Koysha | 93.7 | 84.0 (79.9, 87.4) | 84.4 | 90 | 99.0 |
| Offa | 92.0 | 85.4 (80.0, 89.5) | 85.6 | 90 | 99.9 |
| Sodo Zuria | 93.8 | 88.4 (84.7, 91.3) | 89.2 | 90 | 99.1 |
| Tebela Town | 93.7 | 75.3 (69.4, 80.4) | 78.5 | 90 | 95.9 |
| Humbo Woreda | 93.2 | 87.02 (82.4, 90.6) | 88.5 | 90 | 98.3 |
| Hobicha | 90.2 | 78.8 (69.5, 85.9) | 79.7 | 90 | 98.9 |
| Bayra Koysha | 97.3 | 89.1 (84.7, 92.4) | 89.3 | 90 | 99.8 |
| Kawo Koysha | 95.2 | 83.9 (77.3, 88.9) | 84.9 | 90 | 98.8 |
| Gesuba Town | 98.0 | 79.6 (74.8, 83.7) | 80.04 | 90 | 99.4 |
| **Total** | **92.7** | **81.5 (80.9–82.0)** | **82.4** | **90** | **98.8** |

**Table 3. Comparison of survey coverages with reported administrative coverages, program reach and the Geshiyaro threshold for PZQ, Wolaita Zone, February 2021.**

| Wereda | Reported coverage (%) | Survey coverage with 95% CI | Program reach | Geshiyaro threshold | Drug acceptance (%) |
|---|---|---|---|---|---|
| Boloso Sore | 92.2 | 83.5 (75.9, 89.0) | 84.3 | 90 | 99.1 |
| Sodo Town | 76.6 | 51.3 (43.7, 58.9) | 52.2 | 90 | 98.3 |
| Diguna Fango | 92.7 | 82.6 (77.0, 87.1) | 83.5 | 90 | 98.9 |
| Abala Abaya | 82.2 | 88.2 (84.0, 91.4) | 89.5 | 90 | 98.6 |
| Kindo Koysha | 92.4 | 84.8 (80.6, 88.2) | 85.1 | 90 | 99.7 |
| Offa | 92.8 | 85.9 (80.4, 90.0) | 86.2 | 90 | 99.9 |
| Sodo Zuria | 93.6 | 88.1 (83.7, 91.4) | 88.9 | 90 | 99.0 |
| Tebela Town | 92.1 | 74.3 (68.4, 79.8) | 77.9 | 90 | 95.6 |
| Humbo Woreda | 92.7 | 86.7 (82.3, 90.1) | 87.2 | 90 | 99.4 |
| Hobicha | 91.7 | 80.2 (70.9, 87.1) | 80.9 | 90 | 99.0 |
| Bayra Koysha | 96.0 | 89.0 (84.6, 92.3) | 89.1 | 90 | 99.9 |
| Kawo Koysha | 94.4 | 84.3 (77.7, 89.2) | 85.6 | 90 | 98.4 |
| Gesuba Town | 96.1 | 79.7 (74.9, 83.7) | 80.1 | 90 | 99.5 |
| **Total** | **91.2** | **81.5 (80.9–82.0)** | **82.4** | **90** | **98.9** |

for PZQ. In contrast, pre-SAC had the lowest survey coverage for ALB (76.1%), and individuals within aged 21 to 35 years had the lowest survey coverage for PZQ (76.3%).

The reported administrative coverage for all studied woredas in the 2021 Geshiyaro MDA campaign was greater than that reported during the CES. The highest discrepancy between the two figures was observed in Sodo Town, Tebela Town and Gesuba Town. The discrepancy between survey and reported administrative coverage was lower in five woredas for ALB and seven for PZQ. No woreda met the Geshiyaro coverage threshold of 90%. However, four of 13 woredas survey coverage was proximal to the threshold (Tables 2 and 3).

### Reasons for not being offered albendazole and praziquantel

The overall program reach among the eligible population for both drugs was 82.4% (95% CI; 81.8–82.9%). This indicates that a significant portion (17.6%) of the individuals were not offered drugs during MDA. The main reason for not achieving the target coverage included participant absenteeism during MDA (6.6% ALB and 6.8% PZQ), drug distributors (HEWs) failed to reach all households (4.7% ALB and 4.8% PZQ), and the individuals unable to hear about the program (1.3% ALB and 1.7% PZQ) (Table 4). The lowest program reach for both ALB and PZQ were observed in Sodo Town, at 53.2% and 52.2%, respectively. Abala Abaya, Sodo Zuria, Humbo and Bayra Koysha reported a high program reach for both ALB and PZQ. The highest program reach for ALB and PZQ were observed in SAC at 88.4% and 87.4%, respectively. In general, individuals within age 21 to 35 had the lowest program reach for both ALB (77.2%) and PZQ (76.9%). The overall, program reach among males and females were not statistically different (*p = 0.9* for both ALB and PZQ).

The overall treatment acceptance (ratio of those who swallowed the drugs amongst those who have been offered drugs) was high for both drugs. Only 1.2% and 1.1% of those offered reported not taking ALB and PZQ respectively.

### Discussion

This CES was used to estimate the survey coverage and compare it with the respective administrative coverage reported by HEW during MDA. This survey was conducted as part of the

**Table 4. Reasons for not being offered the drugs, Wolaita Zone, February 2021.**

| Reason not offered drugs | ALB, n (%) | PZQ, n (%) |
|---|---|---|
| Underage | 164 (0.87) | 42 (0.24) |
| Pregnant | 247 (1.31) | 251 (1.45) |
| Breastfeeding | 17 (0.09) | 15 (0.09) |
| Too sick | 145 (0.77) | 139 (0.80) |
| Absent | 1254 (6.64) | 1180 (6.82) |
| Not heard about programme | 252 (1.33) | 287 (1.66) |
| Drugs finished/ran out | 40 (0.21) | 19 (0.11) |
| HEW did not come | 887 (4.69) | 825 (4.77) |
| Other | 150 (0.79) | 133 (0.77) |

evaluation activities implemented by the currently ongoing Geshiyaro project [14]. Reasons for not achieving the desired coverage target of 90% for the Geshiyaro project were identified.

The overall survey coverage across all studied woredas as reported by the CES was 81.5% for both ALB and PZQ. The reported administrative coverage by HEW across all studied woredas was greater than the survey coverage reported here; whereby majority of the woredas reported above 90% coverage. The highest discrepancy between the two figures were observed in Sodo Town, Tebela Town and Gesuba Town. This indicates a problem with reporting system employed by the drug distributors (HEW) during the MDA campaign. The drug distributors may be erroneously reporting the ingestion of drugs. Prior to the next round of MDA, MoH should note the errors raised by the CES, and specific geographies that require closer attention in order to improve their reporting system and thus the validity of their results. Adequate training and supervision covering the tallying of drug distribution should be given to the drug distributors and supervisors. It is also important to motivate the drug distributors to have a better reporting system. Alignment of denominators used to calculate the eligible population between the MoH-lead administrative metrics, and the CES should be undertaken. Discrepancies between the reported administrative and survey coverage such as those noted in the study have been similarly reported in Ethiopia [16].

The coverage figures were lower in five of 13 woredas for ALB and seven of 13 for PZQ between the survey and administrative reports, respectively. According to WHO, the estimates obtained from these two reports are considered to have a low discrepancy or accepted as similar, if the reported administrative coverage lies within the 95% CI of the survey coverage or is within +/- 10 percentage points of the survey coverage [13]. (Detailed validation interpretation adapted from the WHO CES can be seen in S1 Appendix).

In all the surveyed woredas, the survey coverage is below the Geshiyaro coverage threshold (≥90% coverage) for both ALB and PZQ, indicating the need to strengthen the MDA campaign further in all woredas during the next MDA round. Of importance, the survey coverage was below 80% in Sodo Town (52.3% ALB and 51.3% PZQ), Gesuba Town (79.6% ALB and 79.7% PZQ) and Tebela Town (75.3% ALB and 74.5% PZQ). This demonstrates that relative to the rural woredas, urban towns require greater programmatic attention. If programmatic actions are not taken in future MDA rounds, this may create a bottle neck for STH and SCH transmission break in Wolaita.

Program reach is classified as the individual is offered the drug or no, irrespective of swallowing. It measures if there is any issue with the supply chain of the drug, performance of the drug distributors and allows the program to highlight areas where there needs additional attention [13]. The overall program reach among the eligible population was 82.4% for both ALB and PZQ. This indicates that there were considerable number of beneficiaries who were

not offered these drugs during the MDA campaign. The main reason for not achieving the target coverage included beneficiaries' absenteeism during MDA, drug distributors failed to go to all households, and the beneficiaries unable to hear about the program. Therefore, during next rounds of the MDA, it is important to address reasons for not being offered the drugs which included strengthening the social mobilization, repeated visit by the drug distributors, motivation of the drug distributors, and enhancing the engagement of MoH staff during the future MDA campaign.

There was minimal drop out between individuals offered drugs, swallowing them. Therefore, it is important to increase program reach, and ensure more individuals are contacted in future rounds, as once contacted they will likely swallow drugs.

## Conclusions

Programmatic actions need to be taken during the next MDA campaign to increase the program reach (eligible population contacted). The study shows that once contacted, there is minimal drop-off from coverage to compliance, highlighting the requirement of future efforts to focus on widening programmatic reach. Improved training and supervision for tallying used by HEW during MDA should be provided, to improve upon the data capture system currently in place.

## Supporting information

**S1 Appendix. Validation interpretation, adapted from WHO coverage evaluation survey, household questionnaire, adapted from WHO coverage evaluation survey [13] and parameter values used for sample size calculation.**
(DOCX)

**S2 Appendix. Minimum data set.**
(XLSX)

## Acknowledgments

We are very much grateful for all individuals who participated and provided their information for the study. We are indebted to all administrators at each level who facilitated the implementation of the current study.

## Author Contributions

**Conceptualization:** Ewnetu Firdawek Liyew, Melkie Chernet, Habtamu Belay, Rosie Maddren, Toby Landeryou, Alison K. Ower, Kalkidan Mekete, Anna E. Phillips.

**Data curation:** Ewnetu Firdawek Liyew, Melkie Chernet, Habtamu Belay.

**Formal analysis:** Ewnetu Firdawek Liyew, Melkie Chernet, Habtamu Belay.

**Funding acquisition:** Ewnetu Firdawek Liyew, Alison K. Ower, Kalkidan Mekete, Anna E. Phillips, Getachew Tollera, Geremew Tasew.

**Investigation:** Ewnetu Firdawek Liyew, Melkie Chernet, Habtamu Belay, Suprabhath Kalahasti, Alison K. Ower, Kalkidan Mekete, Anna E. Phillips.

**Methodology:** Ewnetu Firdawek Liyew, Melkie Chernet, Habtamu Belay, Rosie Maddren, Toby Landeryou, Suprabhath Kalahasti, Alison K. Ower, Kalkidan Mekete, Anna E. Phillips, Zelalem Mehari.

**Project administration:** Ewnetu Firdawek Liyew, Melkie Chernet, Habtamu Belay, Anna E. Phillips, Ufaysa Anjulo, Adugna Tamiru, Bokretsion Gidey, Getachew Tollera, Geremew Tasew.

**Resources:** Ewnetu Firdawek Liyew, Anna E. Phillips, Ufaysa Anjulo, Bokretsion Gidey.

**Software:** Ewnetu Firdawek Liyew, Melkie Chernet, Habtamu Belay.

**Supervision:** Ewnetu Firdawek Liyew, Melkie Chernet, Habtamu Belay, Ufaysa Anjulo, Tujuba Endrias, Birhan Mengistu, Geremew Tasew.

**Validation:** Ewnetu Firdawek Liyew, Melkie Chernet, Habtamu Belay, Rosie Maddren, Toby Landeryou, Alison K. Ower.

**Visualization:** Ewnetu Firdawek Liyew, Melkie Chernet.

**Writing – original draft:** Ewnetu Firdawek Liyew, Melkie Chernet.

**Writing – review & editing:** Ewnetu Firdawek Liyew, Melkie Chernet, Habtamu Belay, Rosie Maddren, Toby Landeryou, Suprabhath Kalahasti, Alison K. Ower, Kalkidan Mekete, Anna E. Phillips, Ufaysa Anjulo, Tujuba Endrias, Adugna Tamiru, Bokretsion Gidey, Zelalem Mehari, Birhan Mengistu, Getachew Tollera, Geremew Tasew.

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
