## [Decision Letter · Decision Letter 0]

5 Oct 2021

PONE-D-21-27754Coverage evaluation surveys following soil-transmitted helminthiasis and schistosomiasis mass drug administration in Wolaita Zone of Ethiopia- The Geshiyaro projectPLOS ONE

Dear Dr. Liyew,

Thank you for submitting your manuscript to PLOS ONE. After careful consideration, we feel that it has merit but does not fully meet PLOS ONE’s publication criteria as it currently stands. Therefore, we invite you to submit a revised version of the manuscript that addresses the points raised during the review process.

I have completed my evaluation of your manuscript. The reviewers recommend reconsideration of your manuscript following major revision. I invite you to resubmit your manuscript after addressing the comments below and in the attached files. 

When revising your manuscript, please consider all issues mentioned in the reviewers' comments carefully: please outline every change made in response to their comments and provide suitable rebuttals for any comments not addressed. Please note that your revised submission may need to be re-reviewed.

We look forward to receiving your revised manuscript.

Kind regards,

Mariya Y Pakharukova, Ph.D., D.Sc.

Academic Editor

PLOS ONE

Journal Requirements:

2. Please amend your current ethics statement to address the following concern: Please explain i) why written consent was not obtained, ii) whether the ethics committees/IRB approved this consent procedure.

3. You indicated that you had ethical approval for your study. In your Methods section, please ensure you have also stated whether you obtained consent from parents or guardians of minors between 10 and 18 years of age included in the study or whether the research ethics committee or IRB specifically waived the need for their consent.

[I have read the journal's policy and the authors of this manuscript have the following competing interests: ZM is the Manager, Evidence Measurement and Evaluation at the Children’s Investment Fund Foundation. BM is the Program Manager at the Children’s Investment Fund Foundation, the project’s funder]. 

Reviewers' comments:

Reviewer's Responses to Questions

**Comments to the Author**

1. Is the manuscript technically sound, and do the data support the conclusions?

Reviewer #1: Yes

Reviewer #2: Yes

2. Has the statistical analysis been performed appropriately and rigorously? 

Reviewer #1: Yes

Reviewer #2: Yes

3. Have the authors made all data underlying the findings in their manuscript fully available?

Reviewer #1: Yes

Reviewer #2: Yes

4. Is the manuscript presented in an intelligible fashion and written in standard English?

Reviewer #1: Yes

Reviewer #2: No

5. Review Comments to the Author

Reviewer #1: I have no further comments on the manuscript. It is well written. I just have a few questions that can be found on the manuscript which is attached in this review. These questions include explaining the problems causing low MDA coverage and how these can possibly be resolved. Another is how motivation of drug distributors can possibly be done. And finally explaining the denominator used in different indicators for coverage used.

Reviewer #2: Comments to the Author

The manuscript describes Coverage evaluation surveys following soil-transmitted helminthiasis and schistosomiasis mass drug administration in Wolaita Zone of Ethiopia

Comment

• What is the eligible population for PC PZQ and ALB in the survey site and what platform according to WHO , please add more information in the introduction?

• In introduction Please add information regarding the PZQ and ALB dose administered. For PZQ please also describe the dose base on age or ?

• how many structure questions have been asked of the community ? is the any question related to adverse event?

• Line 200 Table 1., Please grouping as age group� 1-4,5-14, 15-20,21-34, 35+

• Figure 1 ( from WHO source and you need permission ). Please add new figure which adaptable for the survey site ( including EAs, subdunit, HH etc)

• The surveyed coverage is 52.3% and the reported coverage is 83.5% ( the coverage compliance gap is high) please more explore in discussion chapter.

• In line 256 “The drug distributors were wrongly reporting on ingestion of the drugs” please change this word because author do not have evidence about this. The reason can be the drug distributor only deliver and no mandatory to ingested immediately (author should add the similar reference)

• In discussion, please elaborate what the interpretation of if reported coverage and survey coverage lower, higher or similar, in other word the converting the results into programmatic action

• The conclusion sections of the abstract and manuscript should not be the same.

• Grammatical error in some parts is needed, author may consult Professional English for editing

6. PLOS authors have the option to publish the peer review history of their article (what does this mean?). If published, this will include your full peer review and any attached files.

Reviewer #1: No

Reviewer #2: No

---

## [Author Response · Author response to Decision Letter 0]

26 Oct 2021

To Mariya Y Pakharukova

Academic Editor

PLOS ONE

Subject: Submission of revised version of a manuscript that addresses the points raised during the review process

Manuscript Number: PONE-D-21-27754

Dear Mariya Y Pakharukova and reviewers, 

Thank you so much for reviewing and forwarding the very valuable comments on our manuscript entitled “Coverage evaluation surveys following soil-transmitted helminthiasis and schistosomiasis mass drug administration in Wolaita Zone of Ethiopia- The Geshiyaro project”. We have considered all of the comments very carefully and have found them to be valuable in enriching the manuscript. Please find below detailed point-by-point responses to the comments made by the Academic Editor and the two reviewers.

A. Academic Editor

S.no Questions raised Responses made 

1. Please ensure that your manuscript meets PLOS ONE's style requirements, including those for file naming. • Thank you so much for your comments. We have carefully read the PLOS ONE style requirement and file naming. Our revised manuscript is now prepared according to these guidelines. 

2. Please amend your current ethics statement to address the following concern: Please explain i) why written consent was not obtained, ii) whether the ethics committees/IRB approved this consent procedure • We accept your comment and the ethics statement has now been amended to address your concern, and the amendments have been made in the following way: ‘’For all non-experimental studies with less invasive procedure, obtaining a verbal consent is the standard requirement of the Institutional Review Board of Ethiopian Public Health Institute (EPHI). Hence, verbal consent was taken from the household after providing a summary of the study purpose in the local dialect, Wolaitina’’ (Page 10, lines 197-200). 

3. You indicated that you had ethical approval for your study. In your Methods section, please ensure you have also stated whether you obtained consent from parents or guardians of minors between 10 and 18 years of age included in the study or whether the research ethics committee or IRB specifically waived the need for their consent.

 • For children younger than 18 years of age, consent was obtained from their primary guardian, and this has been explained under the Ethics section of the revised manuscript (Page 10, lines 200-201). 

[I have read the journal's policy and the authors of this manuscript have the following competing interests: ZM is the Manager, Evidence Measurement and Evaluation at the Children’s Investment Fund Foundation. BM is the Program Manager at the Children’s Investment Fund Foundation, the project’s funder]. 

Please confirm that this does not alter your adherence to all PLOS ONE policies on sharing data and materials, by including the following statement: "This does not alter our adherence to PLOS ONE policies on sharing data and materials.” (as detailed online in our guide for authors http://journals.plos.org/plosone/s/competing-interests). If there are restrictions on sharing of data and/or materials, please state these. Please note that we cannot proceed with consideration of your article until this information has been declared. Please include your updated Competing Interests statement in your cover letter; we will change the online submission form on your behalf.

 • Per the Academic Editor recommendation, we have now updated the competing interest statement in the following way. “I have read the journal's policy and the authors of this manuscript have the following competing interests: ZM is the Manager, Evidence Measurement and Evaluation at the Children’s Investment Fund Foundation. BM is the Program Manager at the Children’s Investment Fund Foundation, the project’s funder. This does not alter our adherence to PLOS ONE policies on sharing data and materials.”

 • We have now uploaded the minimum data set as a supporting information file (File name: S4 Appendix: Minimum Data Set) (Page 19, line 375).

B. Reviewers' comments

Reviewer # 1

S.no Questions raised Responses made 

1. I have no further comments on the manuscript. It is well written. I just have a few questions that can be found on the manuscript which is attached in this review. 

A. Can problems related to the reporting system be identified already and so be targeted for correction or solution?

B. How motivation of drug distributors can possibly be done.

C. Explain more on checking the denominator used to calculate the reported administrative coverage 

 • Thank you so much for the valuable comments. At this stage, it is difficult to identify problems related to the reporting system (as it is beyond the scope of our manuscript). Based on the findings of our study, we indicated that there are problems related to the reporting system. We recommended that Data Quality Self-Assessment (DQSA) need to be done by the Ministry of Health (MoH), MDA implementing partner, to identify problems related to the reporting system. Therefore, unless the DQSA is done, it is difficult to identify all those problems related to the reporting system. DQSA allows MoH to better understand where the inaccuracy in reporting system is taking place. Problems in reporting system might be related to the overall MDA program data management system employed by the MoH or the data collection/review process happening at different level of the reporting system. Therefore, developing an action plan prior to next MDA rounds should be done depending on the findings of the DQSA. In order to make the statement more clear, we have now revised the statement on the reporting system in the following way: 

‘’ Prior to the next round of MDA, MoH should note the errors raised by the CES, and specific geographies that require closer attention in order to improve their reporting system and thus the validity of their results’’.

• Motivation of the drug distributors can be achieved via the following key activities to be implemented by the MoH

Providing recognition and award for high performing drug distributors following the MDA campaign. 

Provision of adequate perdiem for drug distributors during the time of MDA campaign. 

Prepare a review meeting following the MDA to discuss on the lessons learnt.

• Denominators used to calculate reported administrative coverage is now better explained in the revised manuscript and read in the following way: 

“Alignment of denominators used to calculate the eligible population between the MoH-lead administrative metrics, and the CES should be undertaken” (page 16, lines 276-277). 

Reviewer # 2

S.no Questions raised Responses made 

1. What is the eligible population for PC PZQ and ALB in the survey site and what platform according to WHO, please add more information in the introduction? 

 • Thank you so much for your comments. The below additional information regarding the eligible population of ALB and PZQ has been added in the introduction section of the revised manuscript (Page 5, lines91-94).

“The eligible population was calculated according to WHO guidelines and is drug specific: individuals aged 1 and older are eligible for ALB, whilst individuals older than 4 years are eligible for PZQ”

2. In introduction Please add information regarding the PZQ and ALB dose administered. For PZQ please also describe the dose based on age or?

 • All eligible community members were offered one dose of 400mg ALB (≥ 2 years old), one bottle syrup of 200 mg ALB ( < 2years old) and 600mg PZQ (>4 years old) administered in a height-dependent dose (1-5 tablets). This information has now been added in the introduction section of the revised manuscript (page 5, lines 94-96).

3. How many structure questions have been asked of the community? Is there any question related to adverse event? 

 • For our study, we have used the questions recommended by the WHO coverage evaluation survey guideline. We have indicated this information in our method section of the manuscript and the detailed questionnaire used in the study can be seen from the Supporting Information 2 (page 19, line 373). Accordingly, the communities were asked about different questions such as adverse events following treatment, source of information for the MDA and motivation to take part in the MDA. 

4. Line 200 Table 1., Please grouping as age group� 1-4,5-14, 15-20,21-34, 35+ 

 • Thank you for your comment and we have now corrected per the reviewer’s suggestion (page 11, line 212).

5. Figure 1 (from WHO source and you need permission). Please add new figure which adaptable for the survey site ( including EAs, sub-unit, HH etc)

 • Thank you for your comment and concern. We have explained in our manuscript that it is adapted from the WHO and cited a reference for this. Actually, we have made a few modifications on the figure to align with the study we conducted. 

6. The surveyed coverage is 52.3% and the reported coverage is 83.5% (the coverage compliance gap is high) please more explore in discussion chapter. 

 • As the reviewer explained, there is a big gap between the survey and reported coverage particularly in 3 Towns (Sodo Town, Tebela Town and Gesuba Town). The highest discrepancy between these figures is an indicative that there was issues with the reporting system employed by the drug distributors. The reason for the observed gap along with possible programmatic recommendations is now better described in the discussion section of the revised manuscript (pages 15 &16, lines 265-279).

7. In line 256 “The drug distributors were wrongly reporting on ingestion of the drugs” please change this word because author do not have evidence about this. The reason can be the drug distributor only deliver and no mandatory to ingested immediately (author should add the similar reference)

 • Per the reviewers comment, we have now changed the wording in the revised manuscript (page 16, lines 270-271).

8. In discussion, please elaborate what the interpretation of if reported coverage and survey coverage lower, higher or similar, in other word the converting the results into programmatic action

 • Comparing survey coverage with reported coverage is used to check whether the data reporting system is working or not. Per these, the interpretation will be done according to the following three scenarios observed in the study. If the reported coverage is higher than the survey coverage, it implies that the routine administrative reporting might be overestimating the true coverage. If the reported coverage is lower than the survey coverage, this is an indication that the routine administrative reporting is likely underestimating true coverage. Finally, if the reported and survey coverages are similar, it indicates that a good reporting system is in place. The detailed interpretation covering all of these three scenarios can be from the Supporting Information 1 (page 19, line 372). 

9. The conclusion sections of the abstract and manuscript should not be the same.

 • Corrected per the reviewers comment (page 3, lines 58-62 and page 18, lines 308-313). 

10. Grammatical error in some parts is needed, author may consult Professional English for editing 

 • The English language has now been copy-edited by a colleagues whose first langue is English.

---

## [Decision Letter · Decision Letter 1]

10 Nov 2021

PONE-D-21-27754R1Coverage evaluation surveys following soil-transmitted helminthiasis and schistosomiasis mass drug administration in Wolaita Zone of Ethiopia- The Geshiyaro projectPLOS ONE

Dear Dr. Liyew,

Thank you for submitting your manuscript to PLOS ONE. After careful consideration, we feel that it has merit but does not fully meet PLOS ONE’s publication criteria as it currently stands. Therefore, we invite you to submit a revised version of the manuscript that addresses the points raised during the review process.

ACADEMIC EDITOR:Thank you for submitting your revised manuscript to PLoS One. Only Reviewer 2 still claimed a number of minor corrections and improvements. The authors are invited to address these and resubmit a further improved revised manuscript accompanied by your detailed response to all comments point-by-point, including a description of the changes made in the revised manuscript. The alterations in the text should be highlighted.

We look forward to receiving your revised manuscript.

Kind regards,

Mariya Y Pakharukova, Ph.D., D.Sc.

Academic Editor

PLOS ONE

Journal Requirements:

Reviewers' comments:

Reviewer's Responses to Questions

**Comments to the Author**

1. If the authors have adequately addressed your comments raised in a previous round of review and you feel that this manuscript is now acceptable for publication, you may indicate that here to bypass the “Comments to the Author” section, enter your conflict of interest statement in the “Confidential to Editor” section, and submit your "Accept" recommendation.

Reviewer #2: All comments have been addressed

2. Is the manuscript technically sound, and do the data support the conclusions?

Reviewer #2: Yes

3. Has the statistical analysis been performed appropriately and rigorously? 

Reviewer #2: Yes

4. Have the authors made all data underlying the findings in their manuscript fully available?

Reviewer #2: Yes

5. Is the manuscript presented in an intelligible fashion and written in standard English?

Reviewer #2: Yes

6. Review Comments to the Author

Reviewer #2: Overall, it looks good now, but some minor changes are required.

Point number 4 : Line 200 Table 1., Please grouping as age group� 1-4,5-14, 15-20,21-34, 35+

Not as I expected Please add Age group at colum above age (see in attachement)

Line 92 Please revised the sentences Mothers in their third trimester are advised

93 against taking ALB or PZQ to ….Mothers in their third trimester are advised not to take ALB or PZQ

7. PLOS authors have the option to publish the peer review history of their article (what does this mean?). If published, this will include your full peer review and any attached files.

Reviewer #2: No

---

## [Author Response · Author response to Decision Letter 1]

14 Nov 2021

To Mariya Y Pakharukova

Academic Editor

PLOS ONE

Subject: Submission of revised version of a manuscript that addresses the points raised during the review process

Manuscript Number: PONE-D-21-27754R2

Dear Mariya Y Pakharukova and reviewer, 

Thank you so much for reviewing our revised manuscript entitled “Coverage evaluation surveys following soil-transmitted helminthiasis and schistosomiasis mass drug administration in Wolaita Zone of Ethiopia- The Geshiyaro project”. Please find below point-by-point responses to the few comments made by the Academic Editor and the second reviewer.

A. Academic Editor

S.no Questions raised Responses made 

1. Please review your reference list to ensure that it is complete and correct. If you have cited papers that have been retracted, please include the rationale for doing so in the manuscript text, or remove these references and replace them with relevant current references. Any changes to the reference list should be mentioned in the rebuttal letter that accompanies your revised manuscript. • Thank you so much. We have used a Reference Manger to cite the references. We haven't used retracted articles in our manuscript. We have made some modification on reference number 11 (WHO. Preventive chemotherapy in human helminthiasis. 2006). This reference has now been changed in the following way in the revised manuscript ( page 19, lines 351-353). 

Crompton, David WT, WHO. Preventive chemotherapy in human helminthiasis : coordinated use of anthelminthic drugs in control interventions : a manual for health professionals and programme managers. Geneva: World Health Organization; 2006.

• We have also deleted reference # 15 (WHO. Preventive chemotherapy in human helminthiasis : coordinated use of anthelminthic drugs in control interventions : a manual for health professionals and programme managers. Geneva: World Health Organization; 2006) as it is similar to reference # 11. 

B. Reviewers' comments

Reviewer # 2

S.no Questions raised Responses made 

1. Point number 4 : Line 200 Table 1., Please grouping as age group 1-4,5-14, 15-20,21-34, 35+. Please add Age group at column above age. • Thank you so much and this has been corrected per the reviewer’s suggestion (page 11, line 211). 

2. Line 92 Please revised the sentences Mothers in their third trimester are advised

93 against taking ALB or PZQ to ….Mothers in their third trimester are advised not to take ALB or PZQ • We accept the comment and revision on these sentences has been made in the revised manuscript (page 5, lines 92-93).

---

## [Editor Report · Decision Letter 2]

16 Nov 2021

Coverage evaluation surveys following soil-transmitted helminthiasis and schistosomiasis mass drug administration in Wolaita Zone of Ethiopia- The Geshiyaro project

PONE-D-21-27754R2

Dear Dr. Liyew,

We’re pleased to inform you that your manuscript has been judged scientifically suitable for publication and will be formally accepted for publication once it meets all outstanding technical requirements.

Kind regards,

Mariya Y Pakharukova, Ph.D., D.Sc.

Academic Editor

PLOS ONE
---

## [Editor Report · Acceptance letter]

1 Dec 2021

PONE-D-21-27754R2 

Coverage evaluation surveys following soil-transmitted helminthiasis and schistosomiasis mass drug administration in Wolaita Zone of Ethiopia- The Geshiyaro project 

Dear Dr. Liyew:

I'm pleased to inform you that your manuscript has been deemed suitable for publication in PLOS ONE. Congratulations! Your manuscript is now with our production department. 

Kind regards, 

on behalf of

Dr. Mariya Y Pakharukova 

Academic Editor

PLOS ONE